# MRI Protocol for Pituitary Assessment in Children with Growth or Puberty Disorders—Is Gadolinium Contrast Administration Actually Needed?

**DOI:** 10.3390/jcm10194598

**Published:** 2021-10-06

**Authors:** Marta Michali-Stolarska, Andrzej Tukiendorf, Anna Zacharzewska-Gondek, Jagoda Jacków-Nowicka, Joanna Chrzanowska, Grzegorz Trybek, Joanna Bladowska

**Affiliations:** 1Department of General and Interventional Radiology and Neuroradiology, Wroclaw Medical University, 50-367 Wroclaw, Poland; marta.michali@gmail.com (M.M.-S.); jagodajackow@yahoo.pl (J.J.-N.); joanna.bladowska@umed.wroc.pl (J.B.); 2Department of Public Health, Wroclaw Medical University, 50-367 Wroclaw, Poland; andrzej.tukiendorf@gmail.com; 3Department of Developmental Endocrinology and Diabetology, Wroclaw Medical University, 50-367 Wroclaw, Poland; chrzanowska.eu@gmail.com; 4Department of Oral Surgery, Pomeranian Medical University in Szczecin, 70-204 Szczecin, Poland; g.trybek@gmail.com

**Keywords:** magnetic resonance imaging, pituitary gland, child, contrast media, gadolinium

## Abstract

The aim of this study was to assess the diagnostic value of non-contrast pituitary MRI in children with growth or puberty disorders (GPDs) and to determine the criteria indicating the necessity to perform post-contrast examination. A retrospective study included re-analysis of 567 contrast-enhanced pituitary MRIs of children treated in a tertiary reference center. Two sets of sequences were created from each MRI examination: Set 1, including common sequences without contrast administration, and Set 2, which included common pre- and post-contrast sequences (conventional MRI examination). The differences in the visibility of pituitary lesions between pairs of sets were statistically analyzed. The overall frequency of Rathke’s cleft cysts was 11.6%, ectopic posterior pituitary 3.5%, and microadenomas 0.9%. Lesions visible without contrast administration accounted for 85% of cases. Lesions not visible before and diagnosed only after contrast injection accounted for only 0.18% of all patients. Statistical analysis showed the advantage of the antero-posterior (AP) pituitary dimension over the other criteria in determining the appropriateness of using contrast in pituitary MRIs. The AP dimension was the most significant factor in logistic regression analysis: OR = 2.23, 95% CI, 1.35–3.71, *p*-value = 0.002, and in ROC analysis: AUC: 72.9% with a cut-off value of 7.5 mm, with sensitivity/specificity rates of 69.2%/73.5%. In most cases, the use of gadolinium-based contrast agent (GBCA) in pituitary MRI in children with GPD is unnecessary. The advantages of GBCA omission include shortening the time of MRI examination and of general anesthesia; saving time for other examinations, thus increasing the availability of MRI for waiting children; and acceleration in their further clinical management.

## 1. Introduction

One of the causes of hormone-related disorders in childhood physical development may be the presence of a lesion in the pituitary gland region. After endocrinological assessment, pituitary changes exclusion is one of the most important elements of diagnostic management of growth and puberty disorders (GPDs) in children [1,2]. The method of choice for pituitary assessment is magnetic resonance imaging (MRI) [2].

Typically, 1.5 T and 3 T MRI scanners are used for imaging of the sellar region, which allows for adequate image quality to be obtained. Of course, due to the higher spatial resolution and higher image quality, 3 T scanners are slowly replacing the 1.5 T ones, but the latter are still used in medical imaging in many centers around the world [3]. The next step in pituitary imaging strategies was the application of various 3-D imaging techniques (e.g., T2 SPACE, GRASP, TWIST or IT-TWIST) to obtain increased sensitivity of multiplanar imaging, thinner layers, and even higher resolution while reducing artifacts [4,5,6,7,8,9,10]. Unfortunately, they have been mainly used thus far on adult patients and most of them require gadolinium administration.

Currently, the MRI protocol of the pituitary gland includes images before and after administration of gadolinium-based contrast agent (GBCA) [2,11], which, according to current knowledge, is not neutral to the human body. Apart from the commonly known gadolinium-related side effects, such as allergic reactions as well as nephrogenic systemic fibrosis (in kidney failure) [12,13], the scientific literature alerts us to the possible accumulation of gadolinium compounds in some brain structures (globus pallidus and dentate nucleus) [14].

These observations are of concern to the medical community and the questions are self-evident: will the developing brains of children have an increased susceptibility to long-term, potentially neurotoxic effects of GBCA deposits [15,16] and is it possible to limit the use of GBCA in pituitary MRI in children [17,18]? Additionally, the omission of GBCA administration would shorten the examination time and the duration of general anesthesia, which is necessary in the youngest patients [2].

Over the past decade, the concern about possible harmful effects of anesthetics and sedatives used in neonates and young children has increased [19,20,21,22]. Studies on animals demonstrate that repeated or prolonged administration of these drugs is potentially neurotoxic and may cause delayed cognitive development [22,23,24].

The aim of the study was to assess the diagnostic value of non-contrast MRI of the pituitary gland in children with GPDs and to determine the criteria indicating the necessity to perform post-contrast examination.

We hypothesized that most of the pituitary MRI examinations in children with GPDs do not require GBCA administration.

To the best of our knowledge, this is the first study focusing on performing MRI of the pituitary gland without gadolinium administration in children.

## 2. Materials and Methods

### 2.1. Study Participants

For this retrospective study, after approval by the local ethics committee (opinion No KB-57/2021), a total of 579 young patients from the Department of Developmental Endocrinology and Diabetology who met several inclusion and exclusion criteria were selected. Inclusion criteria were as follows: contrast-enhanced pituitary MRI examination carried out in our radiology department, age below 18 years, and growth or puberty disorders. Exclusion criteria were inadequate MRI due to artifacts (*n* = 1) or examination with only native scans (*n* = 11). Consecutive pituitary MRIs of 567 children between January 2007 and December 2020 were retrospectively reevaluated (Table 1).

From each pituitary MRI examination, two sets of different sequences were created, where Set 1 was the reference (“control”) group for Set 2, which in turn constituted a notional study group. Set 1 included common sequences without contrast administration and Set 2 consisted of common pre- and post-contrast sequences.

The patients were deliberately not divided into age groups, because they were individuals with hormonal disorders and in their case, according to our observations (the head of the project has 20 years of experience in pituitary imaging) and reports of other research centers [25,26,27], the dimensions of the pituitary gland are not directly proportional to age, which is in a healthy population.

### 2.2. Technical Details and Image Analysis

The MR examinations were carried out using two devices: GE 1.5 T MR Signa HDx (*n* = 425) and 3 T Philips Ingenia (*n* = 142), using 16-channel coils dedicated to the head and neck area. 

T1 weighted images (T1-WI) and T2 weighted images (T2-WI) were taken in the coronal and sagittal planes using thin 3 mm slices before and after intravenous administration of the macrocyclic GBCA. The contrast dose was 0.1 mmol/kg body weight (0.2 mL/kg BW). General anesthesia was additionally used in younger patients (below 7 years) to exclude movement artifacts. 

The size (gland volume and three dimensions) and the morphology of the pituitary gland, possible presence and then location, signal pattern (SP), and potential occurrence of contrast enhancement of focal lesions were retrospectively reassessed on the GE ADW 4.6 as well as the Philips IntelliSpace workstations.

### 2.3. Statistical Analysis

The computation was performed in the R statistical platform [28]. The receiver operating characteristic (ROC) curve analysis was performed using the “pROC” R package [29].

For the purposes of statistical analysis, from each pituitary MRI examination, two sets of different sequences (Figure 1) were created: Set 1, which included common sequences acquired without contrast administration, and Set 2 consisting of common pre- and post-contrast sequences (equivalent to a conventional pituitary MRI examination). Namely, Set 1 consisted of non-contrast sequences: coronal T1-WI, coronal T2-WI, sagittal T1-WI, and sagittal T2-WI. Set 2, in turn, included the same non-contrast sequences: coronal T1-WI, coronal T2-WI, sagittal T1-WI, and sagittal T2-WI, and additional post-contrast coronal and sagittal T1-WI images.

Each patient was assigned a pair of sets (Set 1 and Set 2), where Set 1 was the reference (“control”) group for Set 2, which in turn constituted a notional study group. Sets were not evaluated in pairs, but each set of a given pair (Set 1 and Set 2) was assessed separately and independently, and the evaluating neuroradiologist was blind to patient data. 

After the sets were assessed by a neuroradiologist, Sets 1 and Sets 2 created from the same MRI examination were paired. The differences in the diagnoses (e.g., visibility of focal lesions) between the sets within each pair: Set 1 (common pituitary MRI sequences without contrast administration) and Set 2 (conventional MRI examination with common pre- and post-contrast sequences), were statistically analyzed.

Such a difference would occur if the focal pituitary lesion diagnosed in the post-contrast Set 2 was invisible in the native Set 1. On the basis of the statistical analysis, it was possible to assess the propriety of using GBCA in children with GPD.

The risk of a difference in diagnosis between the sets in each pair was statistically evaluated using logistic regression and expressed by the classic odds ratio (OR) based on the gathered clinical material. The analysis covered such risk factors as age, sex, puberty, pituitary geometrical dimensions (including antero-posterior (AP), transverse (TR), and cranio-caudal dimension (CC)), gland volume, and signal pattern (SP) of the identified focal lesions.

The cut-offs for the presumed risk factors were estimated using ROC analysis, to distinguish between the negative effect (which is the unnecessary use of GBCA) and positive effect (correct GBCA administration) of the post-contrast examination in conventional pituitary MRI.

The study was conducted according to the guidelines of the Declaration of Helsinki, and approved by the Wroclaw Medical University Ethics Committee for conducting research involving humans (permission No KB-57/2021).

## 3. Results

### 3.1. Patient Characteristics

Patient characteristics are shown in Table 1. Contrast-enhanced pituitary MRI examinations of 567 children (boys/girls, 308/259; mean age ± standard deviation (SD), 6.99 ± 2.59 years; range, 0.9–17.4 years; clinical diagnosis: short stature *n* = 509 [boys/girls, 294/215], gigantism *n* = 5 [3/2], precocious puberty *n* = 54 [11/43]) were retrospectively reevaluated.

### 3.2. The Frequency of Focal Lesions

In our study, Rathke’s cleft cyst (RCC) appeared in 66 patients (11.6%), ectopic posterior pituitary (EPP) was visible in 20 patients (3.5%), and clinically confirmed pituitary stalk interruption syndrome (PSIS) and microadenoma (MA) was present in 5 children (0.9%). Additionally, in Sets 1 (sequences without GBCA administration), they were already visible in 53, 20, and 4 cases for RCC, EPP, and MA, respectively; that is, in about 80% of cases in both RCC and MA and 100% of EPP.

When assessing the pituitary gland for the presence of a tumor, in Sets 2 (conventional pituitary MRIs, which included sequences after GBCA administration) compared to native Set 1, the result changed significantly in only one patient (0.18% of all patients), providing the diagnosis of 1 MA which was not visible on non-contrast MRI (Set 1). Moreover, Set 2 also showed 13 new benign RCCs.

### 3.3. Pituitary Dimensions

Logistic regression highlighted the advantage of geometric risk factors over others in determining the need for gadolinium contrast in pituitary MR in children with GPD. Logistic regression revealed that only the AP dimension of the pituitary gland had statistically significant (*p* < 0.05) chances of determining when the omission of GBCA administration is incorrect, with the following results: OR = 2.23, 95% CI, 1.35 to 3.71, *p*-value = 0.002 (Table 2), while the CC dimension and the gland volume were on the border of statistical significance (*p* < 0.1).

The forest plot (Figure 2) based on logistic regression analysis illustrates the advantage of the AP dimension, which, unlike other factors, does not cross the line of no difference (1.0 value), and is therefore statistically significant.

From the ROC analysis for the AP dimension (shown in Figure 3), with the following results: AUC:72.9% (58.0–87.8%), cut-off value 7.5 mm, sensitivity/specificity rates: 69.2%/73.5%, it can be established that the AP dimension is a statistically significant (AUC lower CI 95% > 50%) predictor of the appropriateness of contrast administration in the diagnosis of pituitary focal lesions in children with GPD. Classification of patients based on the AP cut-off = 7.5 mm indicates the proper decision to administer contrast in approximately (AUC∼73%=) 3 out of 4 patients. Furthermore, ROC analysis of the AP dimension shows the best fitted combination of sensitivity (69%) and specificity (74%) rates.

The statistical interpretation of the ROC analyses shown in Figure 4 and Figure 5 is analogous. Both the CC dimension and the gland volume according to the ROC analysis are statistically significant predictors of the appropriateness of contrast administration in the diagnostic management of the discussed group of children. However, an assessment of the AUC for these risk factors (AUC∼66%/69% for CC/Volume—respectively) is less favorable than AUC (∼73%) for the AP dimension. The ROC curves, together with supplementary statistical estimates, are displayed graphically in Figure 3, Figure 4 and Figure 5 and summarized in Table 3.

### 3.4. Signal Intensity of the Lesion

Statistical analysis showed that the signal pattern (SP) of pituitary lesions in the prediction of accurate contrast administration in the diagnosis of children with GPDs is not statistically significant (*p* > 0.05). Furthermore, our observations showed that the SPs of such lesions are varied.

## 4. Discussion

Focal lesions of the pituitary gland in the population of healthy children are unusual, and although in children with hormonal disorders this is also a rare phenomenon, it is more common than in healthy individuals [1].

It has been assumed that the most common suprasellar neoplasm in children is craniopharyngioma; however, the original study proving this statement [30], and cited (unfortunately not always with due diligence) many times later [31,32,33], was conducted in a 20-year period, from the 1970s to the 1990s, when the technical capabilities of diagnostic equipment were much lower, and the study group included children with symptoms suggesting a tumor in this area. 

A more recent paper, from 2019 [34], when the first-line study for diagnosis in the suprasellar region is already 1.5 or 3 Tesla MRI, showed the frequency of focal lesions in this region in children somewhat differently. The authors tried to carefully select the study group to suit the population as much as possible, excluding patients with compression symptoms (also those with hormonal disorders) of the suprasellar area, who, in older, previously published research, formed the basis of the study. In this research on children, with almost 16,000 patients, Souteiro et al. examined the occurrence of all pituitary abnormalities with an incidence rate of 257/100,000 patients [34]. The results of these authors indicated that the pathology frequency was as follows: RCC in 0.03%, EPP in 0.02%, adenomas in 0.04%, and craniopharyngiomas in only 0.006% of all patients.

In our study of 567 pediatric patients with GPDs, three types of focal pituitary lesions were found: RCC, EPP, and MA (11.6%, 3.5%, and 0.9% of all patients, respectively). Moreover, in our research, no craniopharyngioma was visualized. The frequency of focal lesions in our patient group is significantly higher than in the cited study on generally healthy children, most likely due to the characteristics of both pediatric populations [34]. Güneş et al. had similar observations of RCC in children with endocrine disorders [2].

Moreover, our observations showed that in common pituitary MRI sequences without contrast administration (Set 1), focal lesions were already visible in about 85% of cases. 

In conventional pituitary MRIs, which included common pre- and post-contrast sequences (Sets 2), compared to Sets 1, the outcome of pituitary MRI assessment changed only in 2.5% of all patients (mostly benign RCCs) and, importantly, only one patient (0.18%) had MA. This means that only in 0.18% of cases did the administration of GBCA significantly change the examination result.

The appearance of RCC, EPP, and MA is almost pathognomonic, thanks to their characteristic location. RCCs are present between the anterior and posterior lobe [35], EPP is most often located on the floor of the third ventricle, at the median eminence [18], and MAs are situated within the anterior lobe [36].

It is well known that RCC is a benign lesion [37], and if it is small and does not cause a mass effect, it should not be of concern to radiologists or clinicians. Furthermore, its location is so distinctive that the GBCA administration in this case is completely unnecessary.

It is also possible to identify EPP without the use of a contrast agent. The so-called “bright spot” visible on T1 weighted non-contrast MR images in the floor of the third ventricle enables identification of EPP without post-contrast examination [18].

The presence of a focal lesion in the anterior pituitary lobe, which corresponds to MA [36], would seemingly indicate the need for gadolinium administration. However, in light of recent studies, incidentalomas should not be considered a contraindication to hormone therapy [1]. Consequently, the identification of MA-like tumors in a non-contrast examination should not change clinical management of pediatric patients with hormonal disorders; therefore, the diagnosis of MA does not need to be confirmed by a post-contrast MRI. Following this lead, one can come to interesting conclusions: even in the case of an incidentaloma not visible in the non-contrast examination, its potential detection after GBCA administration does not affect the clinical management of children with hormonal disorders. Of course, we should not generalize, and the profit and loss account should be considered on a case-by-case basis.

As mentioned above, the SP of EPP is very characteristic [18], while SPs of RCC and MA depend on their content [38,39]. Furthermore, as our study found, the latter can be very diverse. In addition, our statistical analysis showed that the SP of pituitary focal lesions is not statistically significant as a predictive factor of the accuracy of contrast administration in the diagnostics of children with GPDs, and thus does not affect the results of post-contrast examination.

Taking into account this fact, combined with typical RCC, EPP, or MA locations and potentially redundant MA-like tumors’ confirmation with GBCA, it seems that after excluding the presence of a pituitary lesion in the native examination, the only factor influencing the decision to administer GBCA (that is, to perform a conventional pituitary MRI examination) in the diagnosis of children with GPDs is the size of the pituitary gland.

The appropriateness of using contrast in pituitary MRIs in children with GPDs can be measured by geometric factors, the advantage of which was demonstrated by statistical analysis (Table 2 and Table 3 and Figure 2, Figure 3, Figure 4 and Figure 5). The ROC analysis showed that the CC dimension and gland volume are statistically significant, but logistic regression defined them as borderline values.

The AP dimension is the most accurate factor assessed as a statistically significant predictor in two independent analyses. Moreover, the ROC analysis itself also indicates the advantage of the AP dimension over the other criteria, as it shows the best fitted combination of sensitivity and specificity rates, which indicates the high ability of this predictor to assess when GBCA use is actually necessary and when the administration of gadolinium should be omitted.

Another argument for the superiority of the AP dimension over others, in particular the cranio-caudal dimension (CC), is the fact that children with precocious puberty usually have an enlarged pituitary gland with a convex upper outline, which automatically increases the CC dimension, and which obviously is not related to the presence of a focal lesion of the gland [26,40].

Combining the results of the statistical analysis on our own material with the abovementioned literature data, we believe that in the practical approach, when assessing the size of the pituitary gland in order to diagnose children with GPD, only the AP dimension should be taken into account.

We hypothesized that administration of GBCA in pituitary MRI is unnecessary in the diagnostics of children with GPDs. According to the ROC analysis (Figure 3), the hypothesis holds true for the small size of the pituitary gland. With the AP dimension above the cut-off value (>7.5 mm), this hypothesis becomes false and GBCA should be administered in such cases.

According to the ROC analysis, 73% of patients with an AP dimension > 7.5 mm should receive GBCA, and although the remaining 27% of patients in this group will receive GBCA unnecessarily, statistical analysis showed that the benefit of conventional examination (with pre- and post-contrast sequences) exceeds the potential losses, and it is certainly a better solution than using GBCA for all individuals without exception.

In children with GPDs, a pituitary-oriented MRI is performed prior to hormonal treatment in order to exclude organic causes of hormonal disorders. The conventional MRI examination protocol includes intravenous administration of GBCA in all diagnosed children, and in younger patients, sedation is used to avoid motor artifacts.

Gadolinium deposition has been a very popular topic in recent years and the current state of knowledge is changing dynamically. The literature is consistent with the occurrence of the accumulation phenomenon of linear GBCA after being repeatedly administered [41], e.g., in the deep structures of the brain [14]. Opinions, however, are divided on the potential accumulation of macrocyclic contrasts, which are commonly considered a safer option and therefore often dedicated to pediatric examinations [13,42,43,44]. Another aspect is the potential association between gadolinium depositions and possible clinical consequences, which has not been confirmed so far [44,45], but further observations of the long-term effects of this matter need to be carried out, especially in children due to the sensitivity of their developing brain and life span [15,16]. At present, no histological changes have been confirmed that would arise as a result of gadolinium deposition in the brain tissue after its multiple administration. However, larger studies were conducted only on the adult population [14,46,47] and the study of children was performed on a small number of patients [16].

However, the clinical significance and potential long-term consequences of administration (especially repeatedly) of gadolinium even in individuals with normal renal function still remain unclear [13,45]. This is evidenced by the position of the European Medicines Agency, which recommends using the lowest sufficient dose of macrocyclic GBCA and only in cases where native scans are insufficient [46]. The U.S. Food and Drug Administration is also interested in this phenomenon and continuing to assess its effects in the human body [48].

In our opinion, GBCA administration should be omitted in most pituitary MRIs in children with GPDs. Therefore, we propose the practical approach for pituitary MRI assessment, which will be helpful even for less experienced radiologists (Figure 6).

Worldwide, similar trends are also emerging in other fields of medicine, where, especially in studies on the pediatric population, scientists are trying to abandon GBCA administration [12,49,50] or at least reduce its dose [17,18].

There are also few studies in the literature indicating a need to reconsider the use of GBCA in pituitary MRI, which is not always necessary for the correct interpretation of the sellar region [50,51].

Perhaps, some of the leading research centers, based on many years of experience, omit gadolinium administration, even in the very first MRI examination of the pituitary gland in unquestionable cases in the diagnosis of children with GPDs. Unfortunately, smaller centers, which may be less experienced in this subject, due to the lack of available literature or current guidelines, will follow EBM, which is in line with good clinical practice, and therefore will administer contrast in every MRI examination of the pituitary gland in the diagnosis of children with GPDs, which we believe might be harmful to children.

As with the harmfulness of gadolinium deposits, scientists are also divided over the toxicity of anesthetics in children [19,20,21,22]. Animal studies have confirmed the neurotoxicity of anesthetics and sedatives on the developing brain tissue [23,24], and while these observations cannot be directly extrapolated to the pediatric population, they are of concern. Due to the specificity of the studied age group, most research is retrospective, and therefore not ideal. Some studies have found an association between exposure to anesthetics and neurobehavioral problems or cognitive impairment in the later stages of development, which is especially noticeable with repeated or prolonged exposure [20,21].

Although several new studies have shown no noticeable neurocognitive impairment in children after short-term exposure to anesthetics [20,22], as with gadolinium deposition, the long-term side effects of such exposure are yet unknown. 

Due to constant concerns about the potential harmfulness of GBCA and the potential risk of general anesthesia in younger children, and in accordance with the results presented above, in our opinion, GBCA administration in unquestionable cases is not only unnecessary but could even be considered an impropriety in the medical decision-making process.

Out of concern for the welfare of the youngest patients, we designed the procedure algorithm (Figure 6) for pituitary MRI assessment in children with GPDs before hormonal therapy.

The suggested MR protocol includes only T1 and T2-WI in the coronal and sagittal planes without contrast administration. Post-contrast examination may be omitted if the native MR examination meets the following criteria:The pituitary gland must not present any focal lesions on the T1 and/or T2 image or the lesion must be in the typical location for RCC, EPP, or MA.The AP dimension of the pituitary gland must be smaller than the cut-off value (7.5 mm).

Obviously, if a suspicious or potentially malignant lesion outside the sellar region is detected, not only should GBCA be administered, but the scope of the examination should also be extended to include a complete MRI protocol of the brain.

The present study has several limitations. First, it has a retrospective design; second, none of our patients underwent surgery, therefore histopathological confirmation of the diagnoses made by MRI was not possible. Additionally, the subgroups of children aged <2 and >12 years were small. 

Despite its limitations, to the best of our knowledge, this is the first study involving an attempt to completely omit GBCA in pituitary MRI in children.

## 5. Conclusions

The results of our research have relevant implications for clinical practice or health policy.

For most children, it will be possible to shorten the examination time by omitting GBCA administration, which in turn has a number of advantages. A briefer examination means a reduced time under general anesthesia in younger children.

Furthermore, reducing the procedure time would increase the number of good-quality (without motor artifacts) examinations performed on conscious young patients, where, for the normal duration of MRI, some would need to be sedated.

Shortening the examination time is also important for health policy due to its economic aspects. Reducing the number of post-contrast examinations will save time, during which more children with GPDs waiting in line might be examined. This aspect is particularly important in countries where access to MRI examinations is limited. This will increase the availability of MRI examinations for waiting children and accelerate their further clinical management.

## Figures and Tables

**Figure 1 jcm-10-04598-f001:**
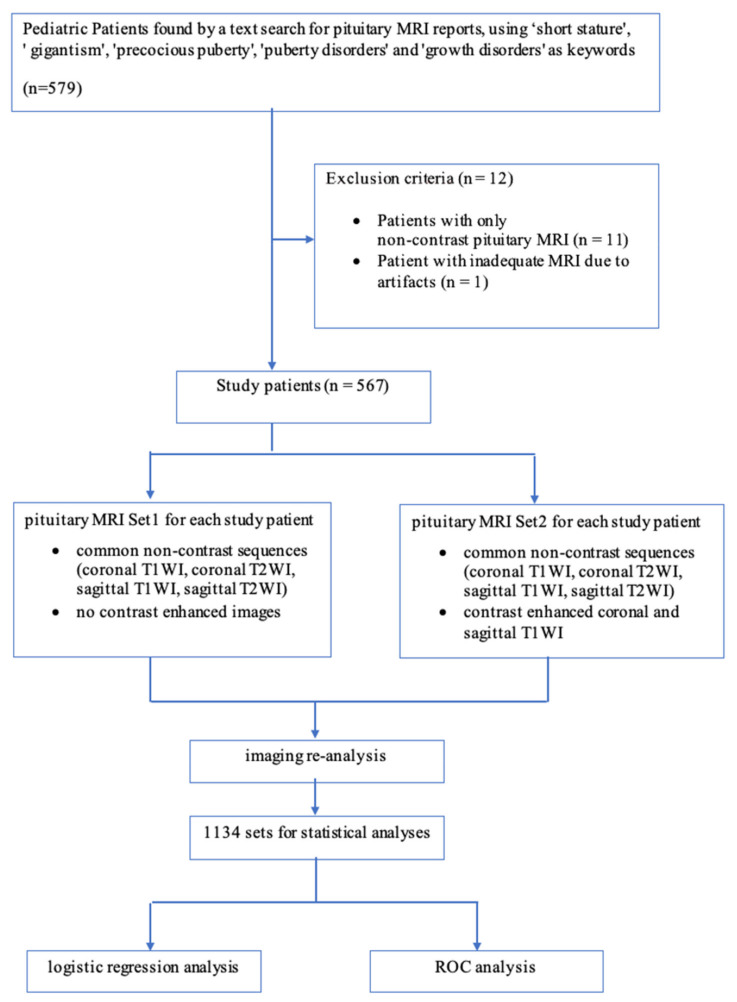
Flowchart of the patient inclusion process and image analysis. T1-WI = T1-weighted imaging, T2-WI = T2-weighted imaging, ROC analysis = Receiver operating characteristic analysis.

**Figure 2 jcm-10-04598-f002:**
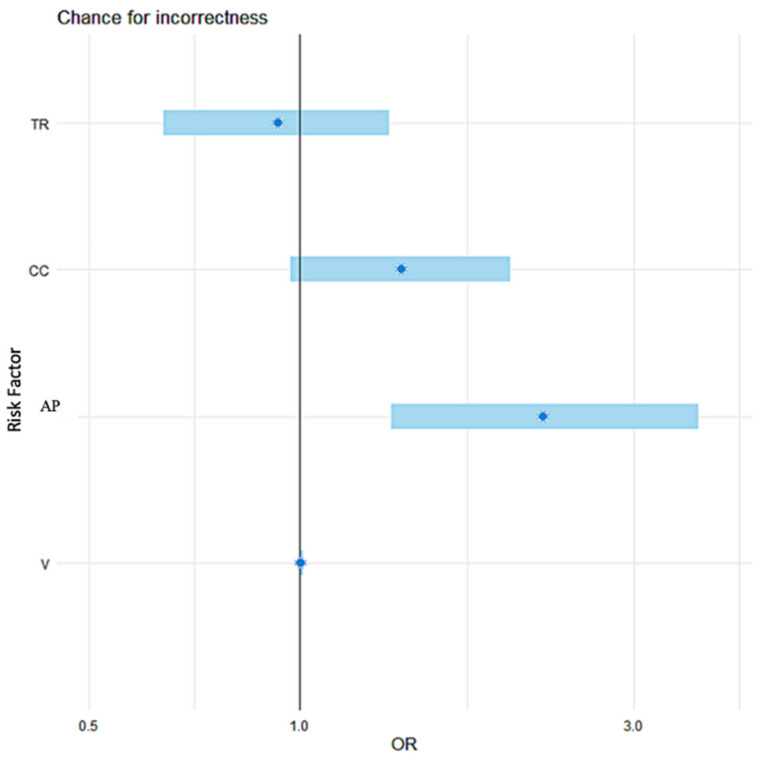
Logistic regression analysis showing advantage of the AP dimension over other geometric criteria determining the appropriateness of using contrast in pituitary MRIs. The forest plot of the odds ratios (OR) of a lack of correct diagnosis of pituitary lesions in a conventional MRI examination (common pre- and post-contrast sequences) includes the following risk factors: antero-posterior pituitary dimension (AP), cranio-caudal dimension (CC), transverse dimension (TR), and gland volume (V). The forest plot based on logistic regression analysis clearly shows that only the AP dimension, which does not cross the line of no difference (1.0 value), is statistically significant in contrast to the other dimensions (CC, TR) and gland volume (V).

**Figure 3 jcm-10-04598-f003:**
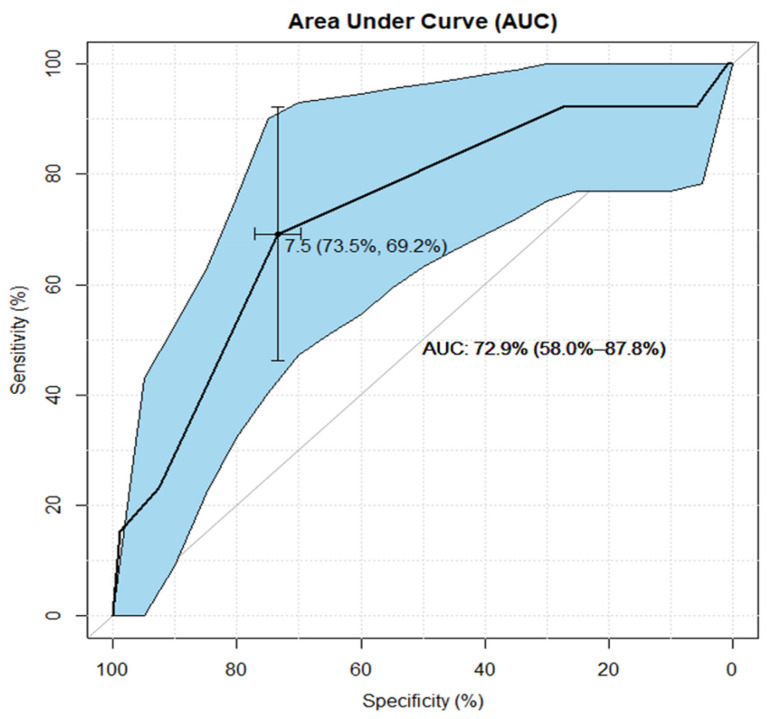
Antero-posterior (AP) dimension in the ROC analysis of the chance for incorrect diagnosis of pituitary lesions in a conventional MRI examination in children with growth or puberty disorders. The receiver operating characteristic (ROC) analysis shows that the antero-posterior (AP) dimension is a statistically significant (area under curve [AUC] lower CI 95% > 50%) predictor of the appropriateness of contrast administration in the diagnosis of pituitary focal lesions in children with growth or puberty disorders (GPDs). Classification of pa-tients based on the AP cut-off = 7.5 mm indicates proper post-contrast diagnosis in (AUC∼73%=) 3 out of 4 patients. Furthermore, ROC analysis of the AP dimension testifies the best fitted combination of sensitivity (69%) and specificity (74%) rates, which reveals the high ability of this predictor to use gadolinium-based contrast agent (GBCA) correctly when required, and the proper omission of GBCA administration when unnecessary.

**Figure 4 jcm-10-04598-f004:**
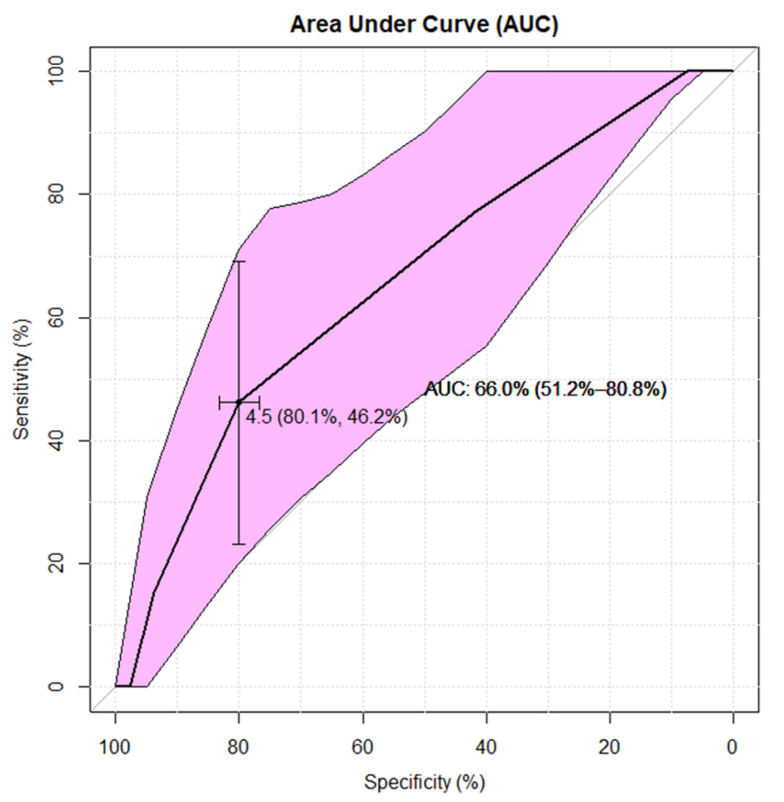
Cranio-caudal (CC) dimension in the ROC analysis of the chance for incorrect diagnosis of pituitary lesions in a conventional MRI examination in children with growth or puberty disorders. The receiver operating characteristic (ROC) analysis shows that the cranio-caudal (CC) dimension is a statistically significant (area under curve [AUC] lower CI 95% > 50%) predictor of the appropriateness of contrast administration in the diagnosis of pituitary focal lesions in our research group. The CC cut-off = 4.5 mm indicates the proper contrast diagnostics in (AUC∼66%=) two out of three patients. Furthermore, ROC analysis of the CC dimension testifies the lower sensitivity rate = 46% than specificity = 80%, which indicates the lower capability of this predictor to use gadolinium-based contrast agent (GBCA) correctly when required.

**Figure 5 jcm-10-04598-f005:**
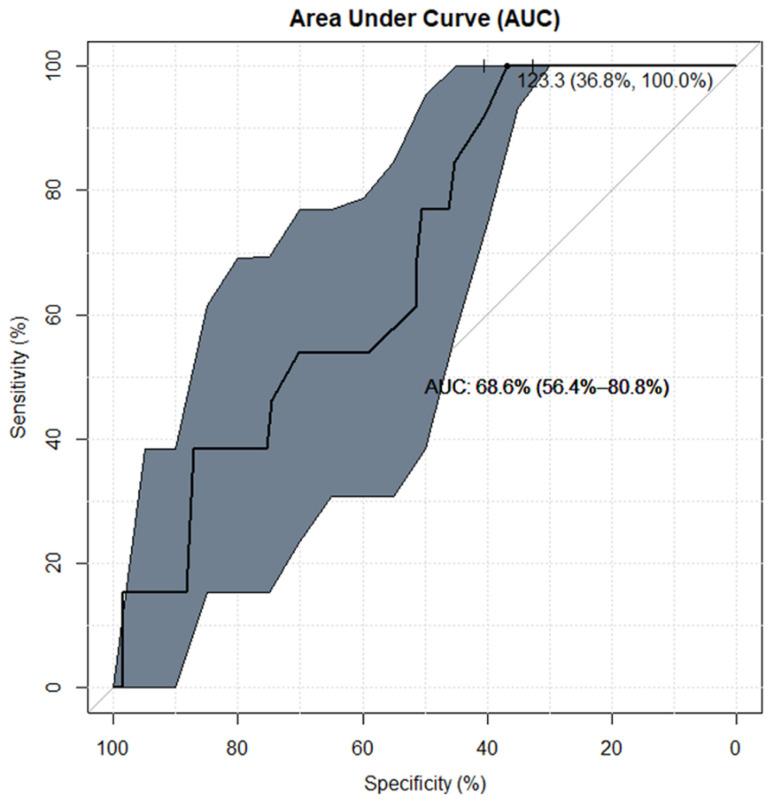
Volume in the ROC analysis of the chance for incorrect diagnosis of pituitary lesions in a conventional MRI examination in children with growth or puberty disorders. The receiver operating characteristic (ROC) analysis shows that pituitary volume (V) is also a statistically significant (area under curve [AUC] lower CI 95% > 50%) predictor of the appropriateness of contrast administration in the diagnosis of pituitary focal lesions in our research group. ROC analysis of the gland volume testifies the lower specificity rate (37%) than sensitivity (100%) and the volume cut-off = 123 mm3 indicates the proper contrast diagnostics in (AUC∼) 69% of patients.

**Figure 6 jcm-10-04598-f006:**
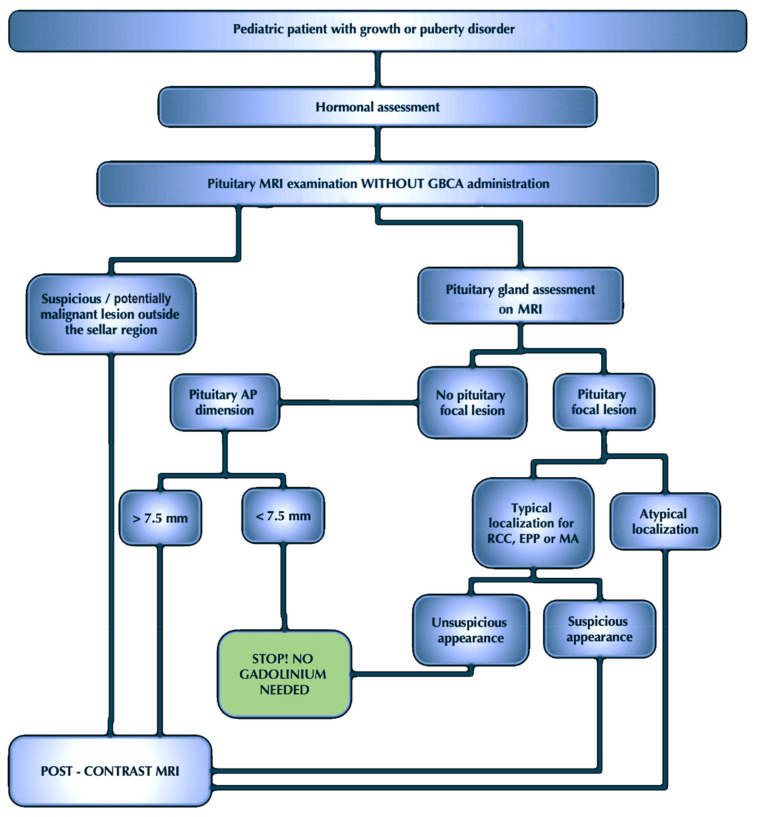
The practical approach for assessment of the pituitary MRI in children with growth or puberty disorders before hormonal therapy. In the new imaging diagnosis algorithm of children with GPDs proposed by us, after endocrinological assessment, patients would have a pituitary MRI examination without GBCA. The results of our study show unequivocally that in most cases, the analysis of native sequences is sufficient to assess the gland to the extent that it allows the patient to qualify for substitution treatment. If the native pituitary MR examination reveals no focal lesion in the parasellar region, and the pituitary AP dimension does not exceed 7.5 mm, the statistical probability of detecting a focal lesion in post-contrast sequences is so small that it is possible to omit gadolinium administration. However, if the AP dimension exceeds 7.5 mm, the examination should be extended by post-contrast sequences. In turn, if a focal lesion in the native MRI is visible, its location and morphology should be carefully assessed. If the lesion location is typical for RCC, EPP, or MA, and the morphology of the focal lesion does not raise suspicions, administration of the contrast agent should also be omitted. However, if the focal lesion has an atypical location, or the location corresponds to RCC, EPP, or MA but its morphology is of concern, in these cases, the pituitary MRI should be extended by post-contrast sequences. Of course, if a suspicious or potentially malignant focal lesion is observed beyond the parasellar region, not only should gadolinium be administered, but the scope of the examination should also be extended, and the entire brain should be visualized.

**Table 1 jcm-10-04598-t001:** Clinical characteristics of the patients. Data are numbers of patients, divided by gender in parentheses: boys/girls.

Characteristic	Value
No. of patients	567
Mean age [years] (± SD)	6.99 ± 2.59
Range of age [years]	0.9–17.4
No. of male patients	294
No. of female patients	259
Clinical diagnosis (M/F):	
- Short stature - Gigantism - Precocious puberty	509 (294/215) 5 (3/2) 54 (11/43)

SD—standard deviation, M—male, F—female.

**Table 2 jcm-10-04598-t002:** Odds ratios (ORs) of the lack of a correct diagnosis of pituitary lesions in a conventional MRI examination (which includes common pre- and post-contrast sequences). Odds ratios (ORs) with 95% confidence intervals (CI 95%) and *p*-values. The table includes tree pituitary dimensions (antero-posterior [AP], cranio-caudal [CC], transverse [TR]) and gland volume [V], where AP is statistically significant (*p* < 0.05), CC and V are on the border of statistical significance (*p* < 0.1), and TR is statistically insignificant.

Risk Factor	OR	CI 95%	*p*-Value
**AP [mm]**	2.23	(1.35, 3.71)	0.002
**CC [mm]**	1.40	(0.97, 2.00)	0.071
**V [mm^3^]**	1.005	(0.999, 1.011)	0.055
**TR [mm]**	0.93	(0.64, 1.34)	0.679

**Table 3 jcm-10-04598-t003:** Summary of the results of the receiver operating characteristic (ROC) analysis.

Risk Factor	AUC	*p*-Value	Cut-Off	Sensitivity (%)	Specificity (%)
**AP [mm]**	0.729 (0.580 − 0.878)	0.002	7.5	69.2	73.5
**CC [mm]**	0.66 (0.512 − 0.808)	0.071	4.5	46.2	80.1
**V [mm^3^]**	0.686 (0.564 − 0.808)	0.055	123.3	100%	36.8

Data in parentheses are 95% CIs. *p*-values are for comparisons of the area under the receiver operating characteristic curve (AUC). Classifications of patients were based on calculated cut-offs, which indicate the proper diagnosis in a conventional pituitary MRI examination (which includes common pre- and post-contrast sequences). The table includes two pituitary dimensions (antero-posterior [AP] and cranio-caudal [CC]) and gland volume [V]. All factors included in the table are statistically significant according to the ROC analysis (AUC lower CI 95% > 50%), but only the AP dimension is statistically significant according to the logistic regression analysis (*p* = 0.002) and also has the best fitted combination of sensitivity (69%) and specificity (74%) rates.

## Data Availability

Data available on request.

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
