# Peer review of "MRI Protocol for Pituitary Assessment in Children with Growth or Puberty Disorders—Is Gadolinium Contrast Administration Actually Needed?"

_jcm, 2021, doi:10.3390/jcm10194598_

Round 1

Reviewer 1 Report

The authors have collected a large series of patients with GPD. The results confirm our own experience.

The topic is not very original but highly relevant and therefore of interest. It is the first large series looking at the issue of administrating Gadolinium in children with growth retardation. Knowing that there are potential complications related to Gadolinium, it is important to become more critical for defining the indications to inject patients. The study confirms what is postulated by many neuroradiologists but the authors now provide evidence of this in their manuscript The text is clearly written and does not need extensive language editing.

It may be good to emphasize the progress in imaging strategies for pituitary pathology (3D & 3T), given the fact that examinations were included between 2007 and 2020.

Reviewer 2 Report

Dear authors, congratulation on your data. It is a nice study with a clear statement to a common every day clinical problem. I like the practical approach as for example the statement that many small findings have no clinical relevance. I have only few questions or ideas. Do you basically see difference between 1.5T and 3 T MRI in regard to this topic. If we consider a smaller center with less experience with MRI interpretation of sella would you suggest the same approach? Is there another MRI sequence or imaging technique which could potentially in the future even improve the diagnostic?
